# ⌨ CoCoPIF: Benchmarking Conversational Coding and Programmatic Instruction Following

## Abstract

Code generation with large language models (LLMs) has become popular to software development, yet existing benchmarks like HumanEval and LBPP focus on single-turn task completion. In real-world scenarios, users often engage in multi-turn interactions, iteratively refining code through instruction-following feedback to meet complex requirements or constraints. Current benchmarks fail to capture the dynamic, instruction-driven nature of such workflows. To address this, we introduce CoCoPIF, a new evaluation pipeline for evaluating LLMs in multi-turn, instruction-following code generation, by emulating real-world interaction data from ShareGPT and problems from LiveCodeBench. Our framework dynamically transforms code problems into multi-turn tasks with verifiable instructions. It features an evaluation protocol that mirrors user-LLM interaction by iteratively refining model outputs through targeted feedback. Furthermore, our assessment approach evaluates both instruction adherence and functional correctness, delivering a reliable measurement of model performance. CoCoPIF reflects practical coding scenarios, providing a tool to assess LLMs in realistic, interactive programming contexts.

## 1 Introduction

Code generation with LLMs has transformed modern software engineering, powerful tools like GitHub Copilot (GitHub, 2021) can assist users in writing, debugging, and optimizing code. Unlike traditional programming tasks, real-world code generation often unfolds through multi-turn interactions, where users iteratively refine their requirements—requesting specific algorithms, modifying code block structures, or enforcing coding style constraints. However, existing evaluation benchmarks, such as HumanEval (Chen et al., 2021) and LBPP (Matton et al., 2024), are designed for single-turn code completion, evaluating models on isolated problem-solving without capturing the dynamic, iterative nature of these interactions. Real-world data from ShareGPT (sha, 2025), a repository of user-LLM dialogues, indicate that over 70% of programming-related sessions involve multiple turns, revealing a notable mismatch between current evaluation frameworks and practical LLM usage. This gap limits their ability to assess LLMs' performance in realistic programming scenarios.

Another critical aspect of LLM performance is instruction following, as we expect LLMs to generalize across a wide range of task constraints in a zero-shot setting without task-specific training. In the context of code generation, strong instruction following capabilities is particularly crucial, as it ensures models can adhere to precise, often complex constraints. Examples may include utilizing specific built-in functions, avoiding certain data structures, or conforming to particular coding styles. However, most existing instruction following benchmarks predominantly focus on generic text generation, such as creative writing, general reasoning and format adaptation. Moreover, as LLMs are trained on ever-larger and more diverse datasets, static evaluation benchmarks face an increasing risk of data contamination. This occurs when models inadvertently memorize test problems or their close variants, undermining the integrity of performance assessments and inflating perceived capabilities. These challenges underscore an urgent need for a new evaluation paradigm that not only mirrors real-world code generation workflows but also incorporates dynamic, contamination-free assessment methods to ensure reliable and robust evaluations of LLMs in programming tasks.

Figure 1: Our framework for interactively evaluating instruction-following data transformation. COCOPIF system generates instructions from a baseline solution, facilitates a feedback-driven evaluation process, and aggregates results from both instruction and runtime checks to produce a comprehensive final output.

To address these challenges, we present CoCoPIF—a novel multi-turn instruction-following framework for code generation evaluation, taking inspiration from real-world interaction data in ShareGPT and code problems from LiveCodeBench (Jain et al., 2024). Our framework is designed to emulate the iterative nature of user-LLM interactions. It incorporates a dynamic data generation pipeline that sources up-to-date problems from LiveCodeBench and transforms them into multi-turn instruction-following tasks, mitigating contamination through the contamination-free nature of LiveCodeBench and an automated, scalable processes (Figure 1). Our benchmark features a diverse set of verifiable instructions, such as replacing `for` loops with `while` loops, adding detailed code comments, or restricting to built-in functions, and enhanced by modern LLMs to produce natural language variations that mirror authentic user queries. This emphasis on verifiability is critical, as it not only addresses the frequent inaccuracies in existing model evaluation but also ensures greater reproducibility of our assessment (Zhou et al., 2023). Moreover, we implement a multi-turn evaluation framework that allows models to iteratively refine their responses based on targeted feedback, reflecting real-world user-LLM workflows. To better assess model performance in real-world scenarios, we implement a dual evaluation metric that integrates instruction comprehension assessment with execution-based functional correctness testing. We present a detailed comparison in the Table 1, highlighting the differences between CoCoPIF and several existing benchmarks.

In a nutshell, our contributions are:

- We introduce a multi-turn instruction-following benchmark tailored for code generation, overcoming the limitations of single-turn frameworks.

- We propose a contamination-free, scalable data generation pipeline that ensures evaluation integrity through verifiable instructions and multi-expression generation.

- We designed an evaluation protocol based on real user interaction patterns, incorporating a dual-dimensional assessment that measures instruction adherence and code functionality.

Table 1: Comparison between CoCoPIF and other code and instruction-following benchmarks.

| Benchmark | Multi-Turn | Contamination-Free | Real Instruction | Dynamic Feedback |
|---|---|---|---|---|
| HumanEval (Chen et al., 2021) | ✗ | ✗ | ✗ | ✗ |
| MT-Bench (Zheng et al., 2023) | ✓ | ✗ | ✗ | ✗ |
| CodeIF (Yan et al., 2025) | ✗ | ✗ | ✗ | ✗ |
| Multi-IF (He et al., 2024) | ✓ | ✗ | ✗ | ✗ |
| LiveCodeBench (Jain et al., 2024) | ✗ | ✓ | ✗ | ✗ |
| CoCoPIF | ✓ | ✓ | ✓ | ✓ |

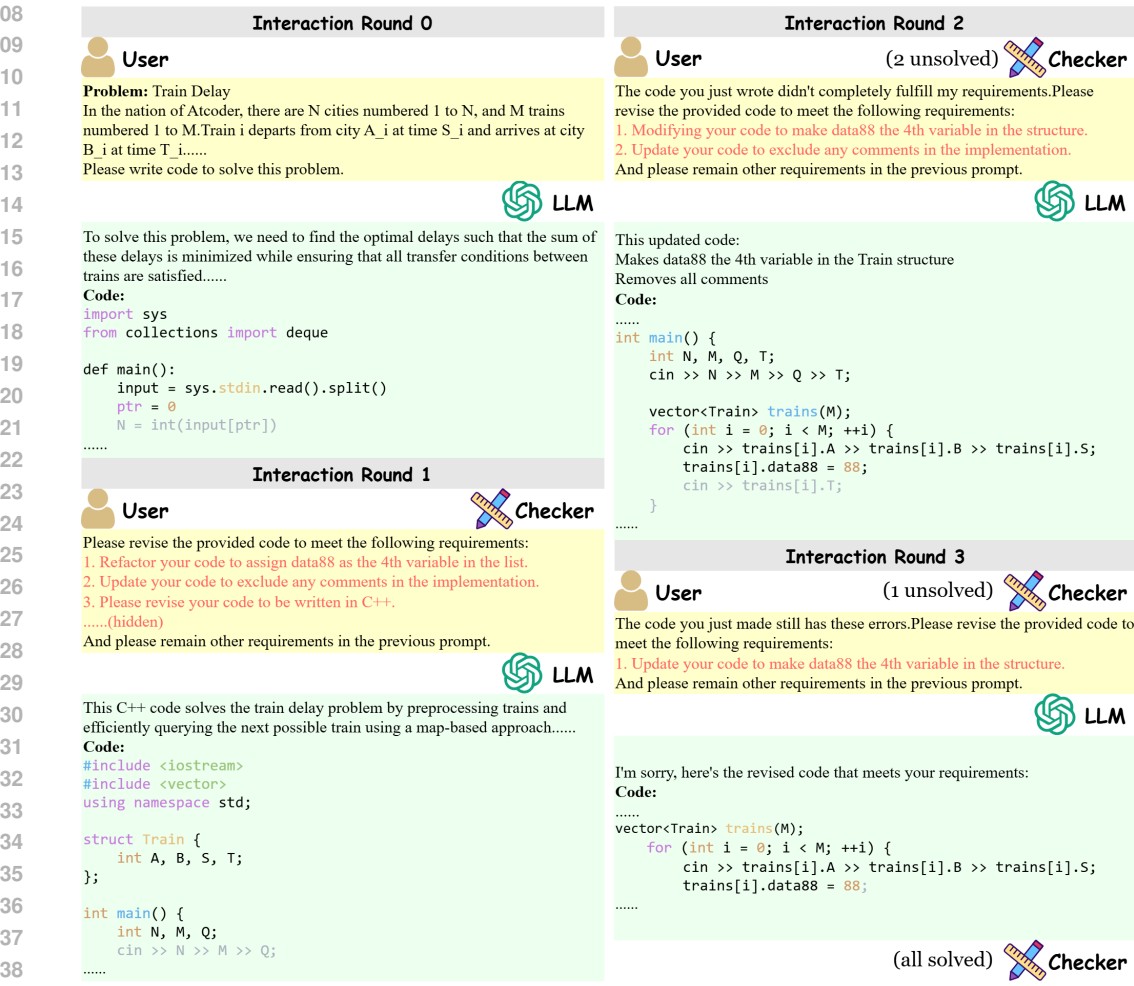

Figure 2: An Interaction example produced by Deepseek-V3. The model's outputs are in the green boxes, and the feedback by our checker is in yellow box. Some details are omitted for clarity.

## 2 RELATED WORK

### 2.1 CODE GENERATION BENCHMARKS

Code generation benchmarks evaluate LLMs' ability to create functional code from natural language prompts. HumanEval uses 164 hand-crafted Python problems with the pass@k metric but is limited by its small scale and Python focus. MBPP (Austin et al., 2021) provides 974 basic Python tasks, but low test coverage hinders its effectiveness for advanced models. APPS (Hendrycks et al., 2021) offers 10,000 diverse problems from platforms like Codeforces, though public sourcing risks data leakage. LBPP (Matton et al., 2024) counters contamination with 161 novel Python problems on complex topics like graph algorithms, but its small size causes variance. LiveCodeBench dynamically collects nearly 1,000 rigorously tested problems from platforms like LeetCode, ensuring freshness and relevance to modern programming practices. Due to its lack of data contamination and its challenging nature, Livecodebench was chosen as the source for the code data in this work.

### 2.2 INSTRUCTION-FOLLOWING BENCHMARKS

Instruction-following benchmarks evaluate LLMs' ability to execute zero-shot tasks with precise adherence to constraints, a critical aspect for real-world applications. FollowEval (Jing et al., 2023) tests bilingual tasks in reasoning and string manipulation with regex-based validation, but its static

design lacks adaptability. FollowBench (Jiang et al., 2024) introduces fine-grained constraints (e.g., style, format) and multi-level testing, combining rule- and model-based evaluations, though model biases remain. IFEval (Zhou et al., 2023) employs 500 prompts with verifiable instructions (e.g., keyword inclusion) for objective assessment, yet it overlooks complex constraint interactions. ComplexBench (Wen et al., 2024) addresses this by testing 1,150 instructions across lexical, semantic, and utility constraints, using rule-augmented LLM evaluation. Realinstruct (Ferraz et al., 2024) leverages authentic user-AI interactions, though ambiguous constraints can affect consistency. For code-specific tasks, CodeIF evaluates multiple languages (e.g., Python, Java) with metrics like Complete Satisfaction Rate, offering nuanced insights into constraint adherence.

### 2.3 MULTI-TURN BENCHMARKS

Extending evaluation to dynamic interactions, multi-turn benchmarks assess LLMs' ability to maintain context and adapt over multiple exchanges, reflecting real-world usage. MT-Bench tests 80 two-turn dialogues across domains like coding and reasoning, using LLMs as judges to reduce subjectivity, but its small scale limits robustness. BotChat (Duan et al., 2024) automates dialogue generation from human seeds, enhancing efficiency, though LLM-based evaluation risks bias. MINT (Wang et al., 2023) simulates tool-augmented problem-solving with 586 instances, supporting varied feedback scenarios, but tool integration adds complexity. MT-Bench-101 (Bai et al., 2024) offers 1,388 dialogues across 13 task types, categorized by perceptivity, adaptability, and interactivity, though its GPT-4-generated (OpenAI et al., 2024) data may introduce bias. Multi-IF extends IFEval to 4,501 multi-turn, multilingual dialogues, ensuring quality through LLM-human hybrid construction, providing a robust framework for complex instruction following. Our work provides a new perspective, evaluating model efficiency in multi-turn scenarios by measuring the number of turns required to complete a code problem, taking into account the unique characteristics of code-related tasks.

## 3 ⌨ CoCoPIF

To evaluate code generation and instruction-following capabilities of LLMs, we introduce CoCoPIF, a high-quality dataset built through a dynamic and automated data curation pipeline. This pipeline leverages contamination-free coding problems, realistic user instructions and automatic transformation methods to produce multi-turn programming tasks that align with practical user needs.

### 3.1 SELECTION OF CODING PROBLEMS

To construct a dataset for evaluating code generation and instruction-following capabilities, selecting appropriate code source data is crucial. Algorithmic problems from platforms like LeetCode, AtCoder, and CodeForces offer numerous test cases for validation, and align with our goals of assessing code quality and instruction-following in LLMs. Thus, we chose algorithmic problems as our evaluation framework, leveraging LiveCodeBench——a dataset designed for code generation. LiveCodeBench gathers fresh, challenging problems from the aforementioned platforms, minimizing contamination through regular updates and our automated data transformation process, ensuring adaptability and dynamism. CoCoPIF comprises of 880 LiveCodeBench problems, each with encrypted test cases (averaging 21.54 per problem, median 14), enabling reliable evaluation while reducing leakage risks.

### 3.2 CURATION OF REALISTIC CODE INSTRUCTIONS

To make our instruction data more aligned with practical scenarios, we choose to select our code instructions from the ShareGPT dataset. ShareGPT, an open-source Chrome extension developed by Steven Tey and Dom Eccleston in 2022, enables users to share ChatGPT conversation logs, hosting over 438,000 diverse dialogues available on HuggingFace. We began by conducting an initial keyword-based screening of the entire ShareGPT conversational dataset to isolate code-related data. From this, we sampled 500 entries for a thorough manual review to better understand user needs and perform a preliminary categorization. We discovered that a large part of this data was either noise or contained instructions that were difficult to verify. Consequently, we utilized GPT-4o-mini to perform a second screening, removing the irrelevant data and applying a more fine-grained classification to the valid entries. The complete classification results are detailed in the Appendix A.1. This process resulted in the instruction categories shown in the Figure 3.

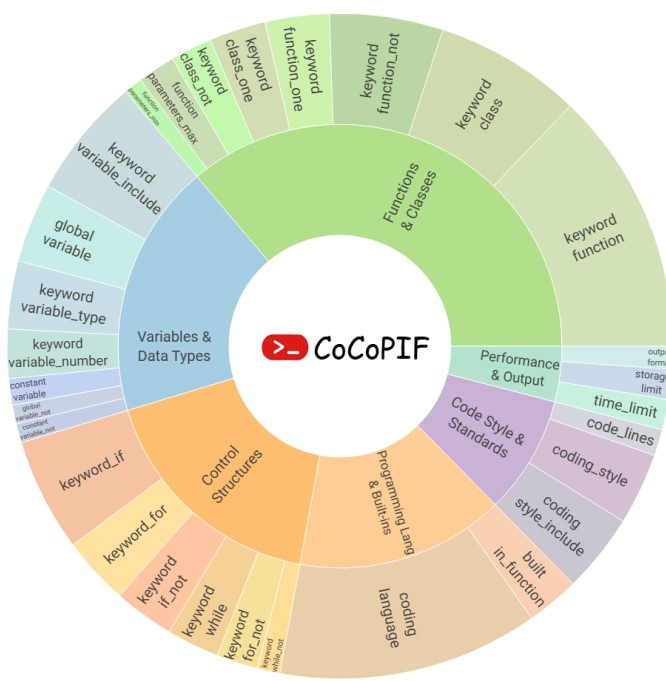

Figure 3: The instruction categories in CoCoPIF. The inner circle shows the major categories, while the outer circle details the fine-grained subcategories for each major one. The arc length corresponds to the proportion of each instruction type in our data.

To ensure the resulting instructions reflect authentic user intent, the distribution of multi-turn instructions was matched to that of ShareGPT functional requests. Starting from the instruction pool distilled from ShareGPT, we used OpenAI's GPT-4o-mini to classify code-related interactions out of 28 368 programming dialogues (prompts in Appendix §A.8).The resulting categories—each appearing $\geq 100$ times, with many surpassing 2 000 occurrences—are summarised in Table 3.

We balanced instruction frequency by weighting categories according to the square root of their occurrence counts to align with real-world scenarios (Figure 3). Our instruction data still face two key challenges to ensure quality. First, we may select conflicting instructions. For example, avoiding for- and while-loops could render some code infeasible. To address this, we identify all conflicting instruction pairs through manual checks and avoid selecting these conflicting instructions during the process. Second, irrelevant instructions require filtering, as some instructions in the dataset proved to be unnecessary for specific problem contexts. For instance, prohibitions against classes became irrelevant when the coding solution inherently did not involve class implementation. To tackle this, we developed a two-stage filtering process, which we illustrate on the left side of Figure 1. We first used GPT-4o-mini to generate baseline solutions, and then eliminated instructions that bore no functional relationship to the actual solution approach.

To incorporate difficulty variation, we randomly assigned 4, 6, or 8 instructions per problem, enabling differentiation by instruction count and problem complexity. This approach ensures a fine-grained difficulty stratification. In our experiment, we also analyzed the impact of different instruction difficulty. For specific results, please refer to the Appendix A.3.

### 3.3 MULTI-TURN TRANSFORMATION

Constructing multi-turn benchmarks requires transforming single-turn source data into realistic multi-turn interactions, which poses a significant challenge. Existing benchmarks adopt two main paradigms. The first, exemplified by Multi-IF, employs a question decomposition approach, breaking complex problems into sequential sub-questions to form multi-turn dialogues. However, this method lacks authenticity, as real users rarely decompose demands proactively; instead, they refine requests based on the model's prior responses, seeking corrections or deeper details. The second paradigm,

represented by MT-Bench, focuses on dependency question construction, ensuring authenticity but requiring labor-intensive data creation, often resulting in limited quantities or fewer interaction turns.

Manual analysis of 500 sampled dialogues from ShareGPT ( A.1) showed that over 50% of conversations exceeding five turns exhibited topic drift, indicating user impatience with a question beyond five turns. Consequently, we set a maximum of five turns as the evaluation threshold to ensure relevance and focus in our dataset. We propose a multi-turn construction pipeline that combines features of existing methods. Unlike Multi-IF, which adds new instructions each turn, our approach presents the full requirement upfront, mimicking real user interactions. After code generation, runtime verification identifies errors, and subsequent turns provide feedback for corrections. This is more realistic than Multi-IF and allows evaluation based on average turns to resolve issues, a practical metric compared to per-turn instruction satisfaction. Unlike the labor-intensive MT-Bench, our fully automated pipeline ensures scalability and precise data control. To help understand our pipeline, we've included an example of the transformed data in the Figure 2 for reference.

When a model fails to resolve an issue within five turns, we evaluate the final output's compliance with the initial instructions, which reflects the practical user tendency to manually refine the output and terminate interaction. Otherwise, we record the number of rounds used to complete the task. This multi-turn evaluation workflow is visualized in the center of Figure 1.

## 4 EXPERIMENTS

Our experiments can be divided into two parts. The first part involves transforming the original programming problem data from LiveCodeBench into instruction-following data. The second part evaluates the number of iterations required to solve the same code instruction-following tasks.

### 4.1 INSTRUCTION-FOLLOWING DATA TRANSFORMATION

Initially, we employ GPT-4o-mini with a temperature setting of 0 to generate deterministic baseline solutions for a given set of coding problems, cost an estimated 0.5 million tokens. These solutions undergo rigorous analysis using Abstract Syntax Tree (AST) parsing to identify and filter out irrelevant or conflicting instructions, ensuring only meaningful instructions remain. Subsequently, we select a variable number of instructions (4, 6, or 8) per problem, guided by weights derived from ShareGPT. To enhance natural language diversity and improve instruction expressiveness, we leverage DeepSeek-v3 to make the selected instructions more natural, resulting in a robust and varied instruction set tailored for effective code generation and evaluation.

### 4.2 EXPERIMENT SETUP

Our experiment simulates a multi-turn dialogue where a user refines requirements, testing LLMs' ability to understand complex instructions, follow detailed constraints, and maintain contextual consistency. It evaluates four key aspects: initial code generation, constraint adherence across iterative versions, multi-turn code refinement based on user's requirements, and the ability to met new requirements while preserving prior constraints.

We tested the following models: GPT-4.1, GPT-4.1-mini, GPT-4o-mini (OpenAI(OpenAI et al., 2024)); Claude-3.7-sonnet, Claude-4-sonnet (Anthropic(Anthropic, 2025)); Gemma-3-27b (Google(Team et al., 2024)), Gemini-2.5-flash (Google(Team et al., 2023)); Qwen-2.5-coder-32b-instruct (Qwen(Hui et al., 2024)), Qwen-2.5-32b-instruct (Qwen(Yang et al., 2024)); Deepseek-v3-0324, Deepseek-v3.1-0821 (Deepseek(DeepSeek-AI et al., 2024)); and Llama-3.3-70b-instruct (Llama(Touvron et al., 2023)). For reproducibility, all models were tested with temperature setting to 0 and a maximum of 5 turns, and their token counts are detailed in the Appendix A.5.

Proprietary models were accessed via OpenRouter API [1], and open-source models were run using the vLLM(Kwon et al., 2023) framework with HuggingFace weights. Our data was generated through multi-turn dialogues between a simulated user and an LLM. For each dialogue, a baseline is first established by having the model solve the problem without any constraints. In up to 5 subsequent turns, the checker extracts code from the prior response, identifies unsolved instructions, and prompts

---

[1] https://openrouter.ai

the model to revise its code accordingly. A dialogue concludes and is saved as a data point once all instructions are successfully solved, or the turn limit is reached. See Figure 1 and Appendix A.4 for details on the generation pipeline.

Table 2: Detailed performance comparison of major LLMs on CoCoPIF.

| Model | Problems Solved | Avg. Turns | IF Pass Rate | Test Case Pass Rate | Pass@1 |
|---|---|---|---|---|---|
| Llama-3.3-70b-instruct | 344 (39.09%) | 3.71 | 53.66% | 12.45% | 7.51% |
| Gemma-3-27b | 258 (29.32%) | 3.22 | 44.37% | 26.53% | 16.08% |
| Qwen2.5-coder-32b-instruct | 391 (44.43%) | **3.16** | 59.03% | 21.15% | 16.51% |
| Qwen2.5-32b-instruct | 286 (32.50%) | 3.83 | 50.34% | 26.74% | 16.99% |
| GPT-4o-mini | 264 (30.00%) | 3.94 | 48.51% | 33.53% | 22.17% |
| Gemini-2.5-flash | 340(38.64%) | 3.57 | 54.86% | 40.56% | 31.95% |
| GPT-4.1-mini | 256 (29.09%) | 3.96 | 48.42% | 54.89% | 42.64% |
| Deepseek-v3 | 352 (40.00%) | 3.48 | 54.64% | 56.45% | 44.78% |
| GPT-4.1 | 319 (36.25%) | 3.67 | 52.43% | 61.78% | 52.12% |
| Claude-3.7-sonnet | 366 (41.59%) | 3.61 | 56.44% | 74.16% | 56.97% |
| Deepseek-v3.1 | 386 (43.86%) | 3.74 | 58.75% | 64.21% | 59.63% |
| Claude-4-sonnet | **428 (48.64%)** | 3.64 | **62.64%** | **76.29%** | **62.72%** |

### 4.3 EVALUATION METRICS AND RESULTS

To better assess model performance in code generation, we adopt a dual-dimensional evaluation framework with the following key metrics:

**Instruction Solving Number**: This measures the number of instruction sets successfully solved by the model within a given problem, directly reflecting its effectiveness. Higher values indicate stronger instruction-following capabilities.

**Average Solving Rounds**: This evaluates efficiency by calculating the average number of interaction rounds needed to solve a problem. Lower values suggest faster and more accurate intent understanding, enhancing user experience.

To enable a more fine-grained comparison of model instruction completion, we also tested the pass rate for instructions in problems, which we call the IF pass rate.

$$\text{IF passrate} = \frac{\text{Instruction}_{\text{solve}}}{\text{Instruction}_{\text{all}}}$$

where $\text{Instruction}_{\text{solve}}$ represents the number of solved instructions, and $\text{Instruction}_{\text{all}}$ represents the total number of instructions in our task.

For code execution performance, in addition to the classic pass@1, we also defined a more granular scoring metric.

$$\text{Testcase passrate} = \frac{n_{\text{pass}}}{n_{\text{all}}}$$

where $n_{\text{all}}$ is the number of all coding testcases, and $n_{\text{pass}}$ is the number of passed testcases. Results are summarized in Table 2 and Figure 5. Meanwhile, we also tested the model's code execution performance for each round of responses. Please refer to the Table 4 for these results.

## 5 ANALYSIS AND DISCUSSION

### 5.1 IMPACT OF INSTRUCTION FOLLOWING ON CODE QUALITY

Figure 4 illustrates the code execution performance of various models across multiple turns, evaluated using the pass@1 metric on CoCoPIF. The radar chart visualizes the performance, where a more uniform shape indicates robustness to instruction demands. Larger models, like Claude-3.7-sonnet, maintain high code quality despite instruction demands, while smaller models, like Gemma-3-27B, show a notable decline in code quality. These results indicate that general-purpose capabilities may help mitigate interference from instruction-following demands.

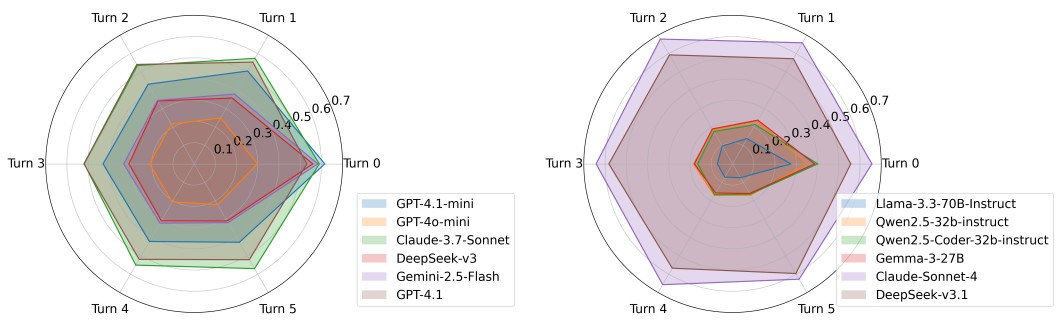

Figure 4: Pass@1 performance degradation over successive conversational turns.

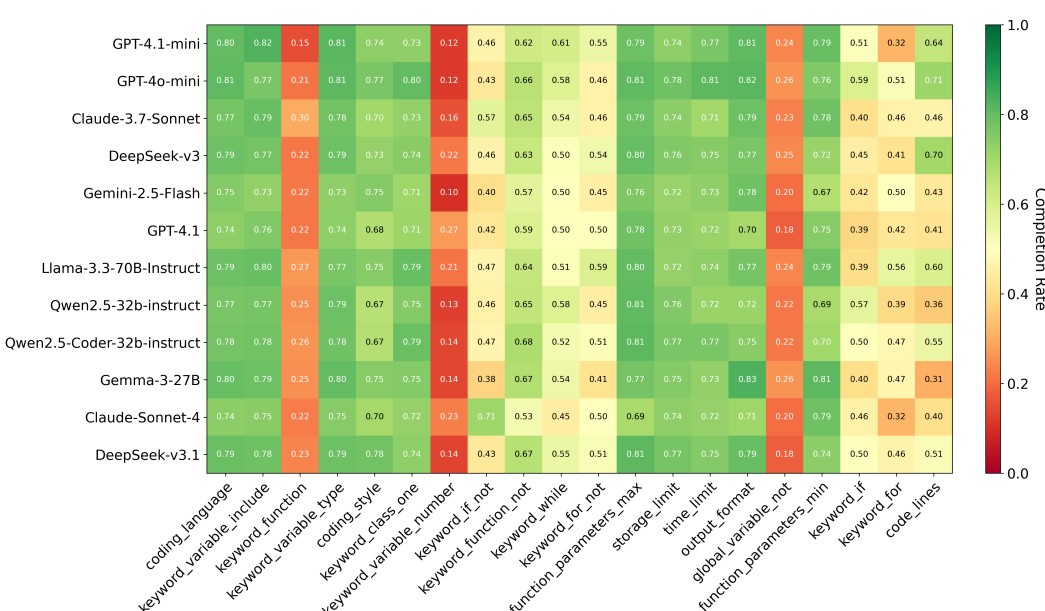

Figure 5: Comparative analysis of instruction completion rates across multiple LLMs.

## 5.2 ANALYSIS ON INSTRUCTION DIFFICULTY

Figure 5 illustrates the varying difficulty of instruction types from ShareGPT data across all models. Simple instructions, such as specifying a programming language, have high completion rates, while complex ones, like specifying the number of functions or the name of the $n$-th variable, have low completion rates. This disparity likely arises because models struggle with precise counting or index-ing, such as identifying the "third variable," which requires robust internal reasoning. Additionally, instructions like specifying function counts demand dynamic state tracking, which is challenging in long code generation. Assigning specific variable names, such as requiring the "second variable to be var_88," further complicates matters by necessitating precise mapping and conflict avoidance.

These challenges highlight deficiencies in precise logical control, state tracking and numerical reasoning. Take DeepSeek-v3 as an example, the input length distribution for the final round averages 4,380 tokens, with extremes reaching 17,500+ tokens. This places the instruction set in the long-context regime, contributing to its difficulty. For detailed information, please refer to the Appendix A.3 for other models' response length distribution.

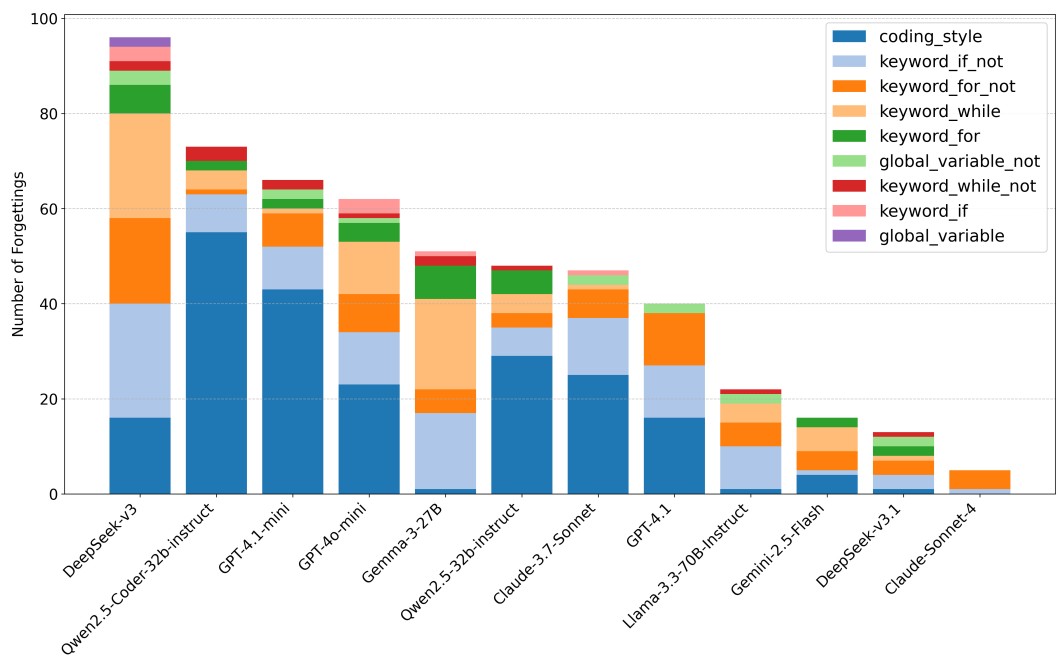

Figure 6: The number and distribution of instruction forgetting phenomena across different models (lower is better).

## 5.3 INSTRUCTION FORGETTING PHENOMENON

The instruction forgetting phenomenon refers to cases where the model output in follow-up rounds ignored some of the instructions stated in the beginning. Take GPT-4.1 as an example, instruction forgetting is prominent for tasks like removing comments and avoiding `if` statements (see Figure 6). Similarly, GPT-4o-mini exhibits similar issues. In contrast, Google's Gemini-2.5-flash and Gemma-3-27b models struggle more with `while` loop instructions but handle comment removal better. This suggests model-specific forgetting patterns, likely tied to biases in pretraining and fine-tuning data. We believe this phenomenon is likely related to the differences in pre-training data adopted by different model families. Further investigation is needed to confirm these findings.

Figure 6 also quantifies instruction forgetting across models, with Deepseek-chat-v3 showing the highest forgetting rate (96 instances) and Claude-4-Sonnet the lowest (4 instances). Simultaneously, we can clearly observe that instruction forgetting is notably reduced in the two newer models, Claude-4-Sonnet and DeepSeek-V3.1, compared to their predecessors. This considerable advancement in instruction adherence strongly suggests that these models have received more targeted fine-tuning for handling complex tasks. From these results, we can observe that the phenomenon of instruction forgetting does not appear to be strongly correlated with the model's code problem-solving ability. In fact, this might be more related to their instruction-following performance.

## 6 CONCLUSION

We introduce CoCoPIF, an evaluation pipeline designed to overcome the limitations of traditional single-turn code evaluations. By simulating realistic, multi-turn programming interactions using a dynamic, contamination-free pipeline with problems from LiveCodeBench and user patterns from ShareGPT, CoCoPIF assesses a large language model's ability to iteratively follow complex instructions while maintaining code functionality. Our experiments reveal that even some powerful models like Claude-3.7-sonnet struggle with precise logical constraints and exhibit a significant "instruction forgetting" phenomenon—weaknesses invisible to conventional benchmarks. CoCoPIF thus provides a more practical framework for evaluating and advancing the capabilities of LLMs in real-world software development scenarios.

## REPRODUCIBILITY STATEMENT

We provide a detailed description of our data processing for ShareGPT and Livecodebench in Section 3 and Appendix A.1. We outline all model settings, including the API versions and their sources in Section 4.3. We also provide our model temperature in Section 4.2 The full list of prompts used in our experiments can be found in the Appendix A.8. To further improve reproducibility, we have also provided an anonymous code repository, available at https://anonymous.4open.science/r/CoCoPIF-E99B.

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

# A APPENDIX

## A.1 SHAREGPT DATA ANALYSIS

Initial analysis revealed a long-tail distribution of conversation turns, with outliers exceeding 1,000 turns due to users switching topics without starting new conversations. To address this, we conducted a statistical analysis of the rounds of interaction between users and models related to code in the ShareGPT data. As explained in the Section 3, after removing interaction data exceeding 10 rounds, the cleaned statistical results are shown in the Figure 7. The results indicate that the vast majority of

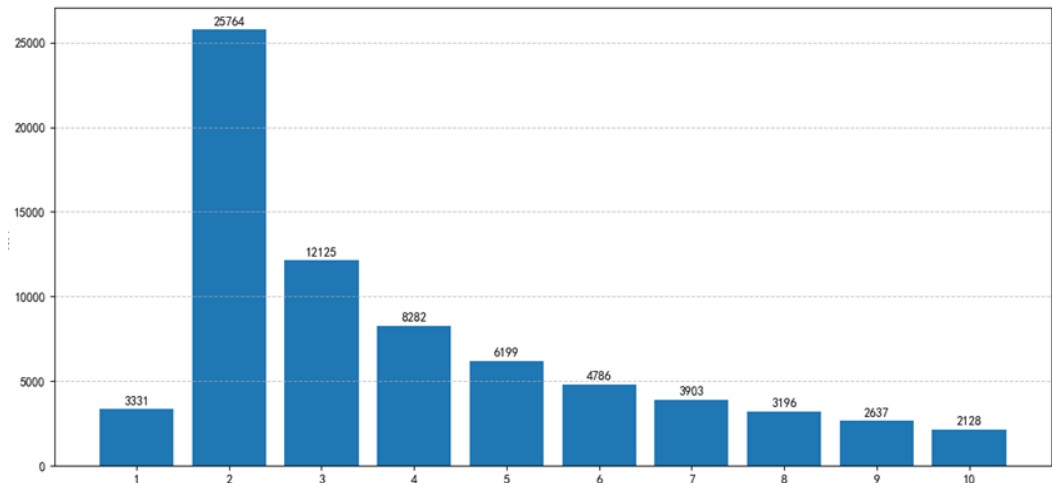

Figure 7: Interaction round distribution.

user interactions involve multi-round dialogues, where users often engage in continuous exchanges to better address complex work-related needs. This finding strongly underscores the importance and necessity of constructing multi-round benchmarks.

Our analysis process can be roughly divided into three steps.

- We performed a statistical analysis on the number of dialogue rounds and used a weighted sampling method to select 500 dialogues for manual annotation. The analysis revealed that over 50% of interactions exceeding 5 rounds exhibited topic drift, where the conversation's later content became unrelated to the initial topic. This usage pattern was identified as erroneous and dialogues exceeding 5 rounds accounted for a relatively small proportion of the total (specific proportions are shown in the histogram). Therefore, we concluded that constructing our evaluation data within 5 rounds is sufficient to ensure data quality.
- Following the process described in Section 3, we utilized GPT-4o-mini for a second screening. This step involved removing irrelevant data and applying a more fine-grained classification to the valid entries.
- After the model completed its classification, we performed a manual sampling check on each category. During this check, we merged or deleted some categories that the model had misclassified, which accounted for a small number, specifically 1.75% of the total. The final classification results are illustrated in Figure 8.

Building on the observation that effective evaluation data can be constrained within five turns, we inspected whether rule-verifiable instructions remain salient in such truncated dialogues. Among the 500 manually annotated dialogues, 312 (62.4%) contained at least one turn that explicitly requested a code-level property that can be checked automatically (e.g., "add unit tests", "make this snippet PEP-8 compliant", "catch the IndexError on line 12"). Crucially, 83% of these verifiable requests appeared in turns 2–4, i.e., after the user had supplied the original snippet and before the fifth-turn threshold where topic drift becomes dominant. This temporal concentration supports our design choice of a five-turn cap: it retains the lion's share of verifiable instructions while excising the noisy, topic-drifted tail. Furthermore, we found that dialogues with exactly three turns exhibited the highest

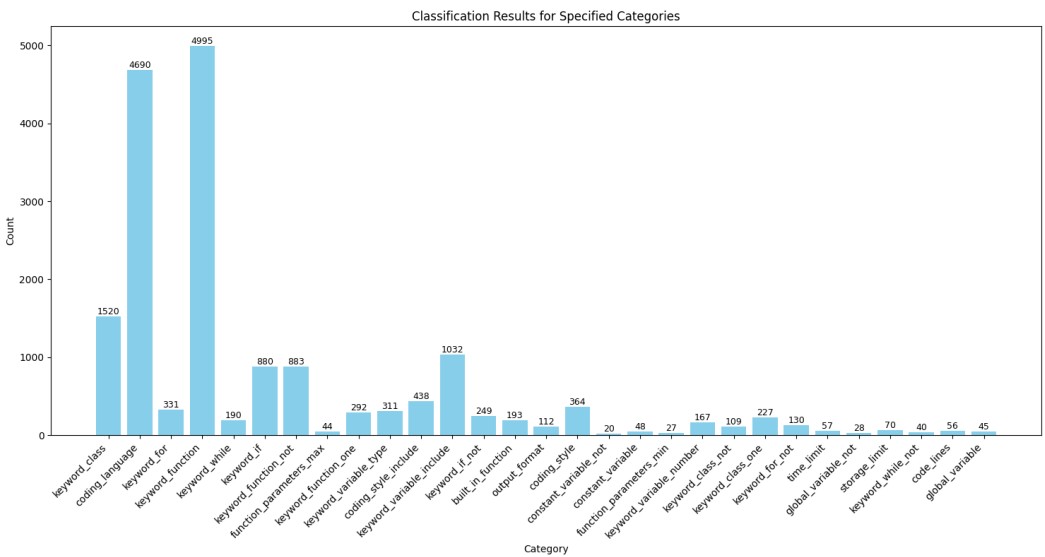

Figure 8: Distribution of different instruction types on ShareGPT.

density of verifiable requests—1.8 per dialogue on average—suggesting that a three-turn format may strike an optimal balance between context richness and annotation cost for future benchmark construction.

## A.2 CODE EXECUTION RESULTS

In our experiment, we also recorded the execution results of the code generated by the model in each round, with the specific outcomes presented in the Table 4.

## A.3 INSTRUCTION DIFFICUTY ANALYSIS

It is insufficient to assess the difficulty of different types of instructions based solely on completion rates. We should also consider completion efficiency, which can be measured by the number of turns required to resolve different instructions. For the average number of rounds used by different models for different instructions, please refer to Figure 9. The instructions selected in this section are all those that have exhibited instances of instruction forgetting.

We also analyzed the token distribution of the total context length in the final-round input across different models to support our argument in the text. The distribution is shown in Figure 10 and Figure 11. It can be observed that some smaller models, such as Llama and Qwen, tend to generate longer outputs, which also reflects their efficiency in solving problems.

## A.4 DETAILS OF EXPERIMENT SETTING

Here, we will elaborate on how we filter out valid instructions based on predefined rules and how we provide feedback to the model during evaluation. Since all our instructions are carefully selected, they can be automatically validated.

Constraint checking combines regular expressions and Abstract Syntax Tree (AST) parsing. For Python code, we use the ast module; for C++ and Java, we use clang and javalang, respectively, ensuring accurate structural analysis. Taking keyword_if as an example, we first categorize the code generated by the model into three types: Python, Java, and C++. After classification, we use regular expressions to remove comments and string literals. Then, within the remaining code, we apply regex checks again to verify the presence of if statements.

Table 3: Categories of coding instructions with examples.

| Category Group | Category | Example |
|---|---|---|
| Variables & Data Types | keyword_variable_include | Modify code to use `var87` as a variable name. |
| | keyword_variable_number | Revise code to position `tab18` as the 4th variable. |
| | keyword_variable_type | Update code to define the 3rd variable as `float`. |
| | constant_variable | Update code to include a `constant` variable. |
| | constant_variable_not | Refactor code to remove `constant` variables. |
| | global_variable | Revise code to use a `global` variable. |
| | global_variable_not | Modify code to avoid `global` variables. |
| Control Structures | keyword_for | Refactor code to include a `for` loop. |
| | keyword_for_not | Update code to exclude `for` loops. |
| | keyword_while | Modify code to include at least one `while` loop. |
| | keyword_while_not | Adjust code to omit `while` loops. |
| | keyword_if | Revise code to include an `if` statement. |
| | keyword_if_not | Modify code to avoid `if` statements. |
| Functions & Classes | keyword_function | Update code to include 3 `functions`. |
| | keyword_function_not | Revise code to exclude `functions`. |
| | keyword_function_one | Adjust your code to include `function`. |
| | keyword_class | Modify code to include 6 `class(es)`. |
| | keyword_class_not | Refactor code to remove `classes`. |
| | keyword_class_one | Refactoring your code to integrate `class`. |
| | function_parameters_max | Limit function parameters to 3. |
| | function_parameters_min | Ensure no function has fewer than 2 parameters. |
| Programming Lang & Built-ins | built_in_function | Restrict function usage to built-in functions. |
| | coding_language | Revise code to be written in Java. |
| Code Style & Standards | coding_style | Adjust code to exclude comments. |
| | coding_style_include | Refactor code to include comments. |
| | code_lines | Revise code to have at most 60 lines. |
| Performance & Output | time_limit | Modify code to run within 2048 ms. |
| | storage_limit | Optimize code to use less than 10240 KB. |
| | output_format | Adjust code to output in { Output } format. |

Table 4: Problem solving rounds performance ($Score_{pass}$).

| Model | Round 1 | Round 2 | Round 3 | Round 4 | Round 5 | Round 6 |
|---|---|---|---|---|---|---|
| GPT-4.1 | 554.88 | 583.16 | 566.05 | 543.65 | 544.75 | 543.63 |
| GPT-4.1-mini | 632.71 | 547.25 | 496.83 | 486.28 | 487.01 | 483.05 |
| GPT-4o-mini | 399.16 | 338.16 | 299.91 | 298.10 | 285.13 | 295.09 |
| Claude-3.7-sonnet | 643.96 | 660.18 | 630.01 | 623.87 | 641.20 | 652.59 |
| Gemma-3-27b | 463.62 | 310.93 | 282.49 | 267.19 | 236.13 | 233.43 |
| Gemini-2.5-flash | 601.83 | 413.95 | 380.26 | 367.21 | 358.85 | 356.89 |
| Qwen2.5-coder-32b-instruct | 459.66 | 267.00 | 206.36 | 195.04 | 189.42 | 186.14 |
| DeepSeek-v3 | 593.68 | 586.23 | 574.78 | 569.39 | 567.51 | 565.05 |
| DeepSeek-v3.1 | 605.68 | 586.23 | 556.78 | 499.39 | 493.51 | 492.77 |
| Llama-3.3-70b-instruct | 346.74 | 177.51 | 134.44 | 116.90 | 107.47 | 109.60 |
| Qwen2.5-32b-instruct | 434.51 | 294.24 | 246.67 | 249.72 | 234.28 | 235.29 |
| Claude-4-sonnet | 693.96 | 680.18 | 678.01 | 676.87 | 672.20 | 671.31 |

To evaluate code functionality, we execute generated code with a 5-second timeout and 64MB memory limit in isolated subprocesses, capturing output, errors, execution time, memory usage, and test case pass rates. To ensure efficient evaluation, we set a 90-second total time limit per code.

For more technical details on other instructions, please refer to our released code.

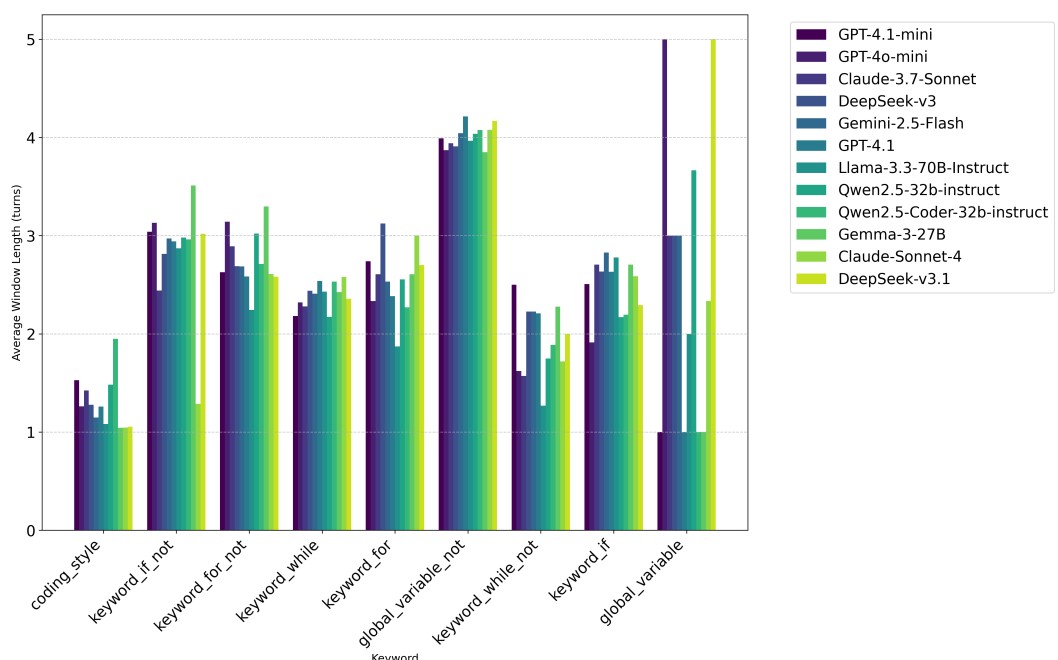

Figure 9: Turns required to resolve different instructions.

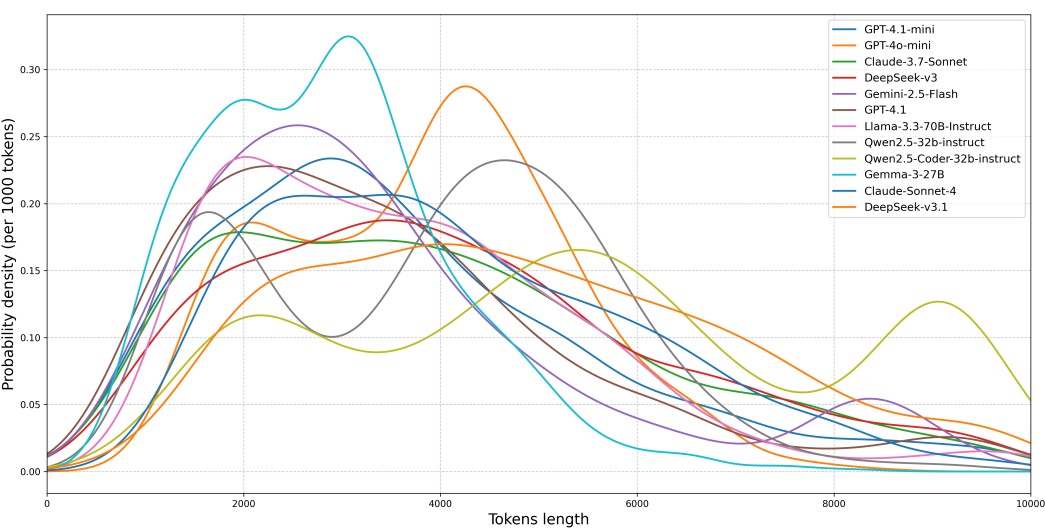

Figure 10: Final round tokens distribution (0-10k).

## A.5 TOKEN COUNT IN EXPERIMENT

Here, to facilitate the reproducibility of the experiments, we provide the total token counts used for all models in our experiments, as shown in the Table 5.

## A.6 THE INFLUENCE OF INSTRUCTIONS AND QUESTION DIFFICULTY ON EACH OTHER

Since our dataset has difficulty classifications for instruction sets (4, 6, and 8 instructions), and the LivecodeBench data itself also includes difficulty classifications for the problems, investigating the mutual influence of these two difficulty dimensions becomes an interesting point.We analyze the code

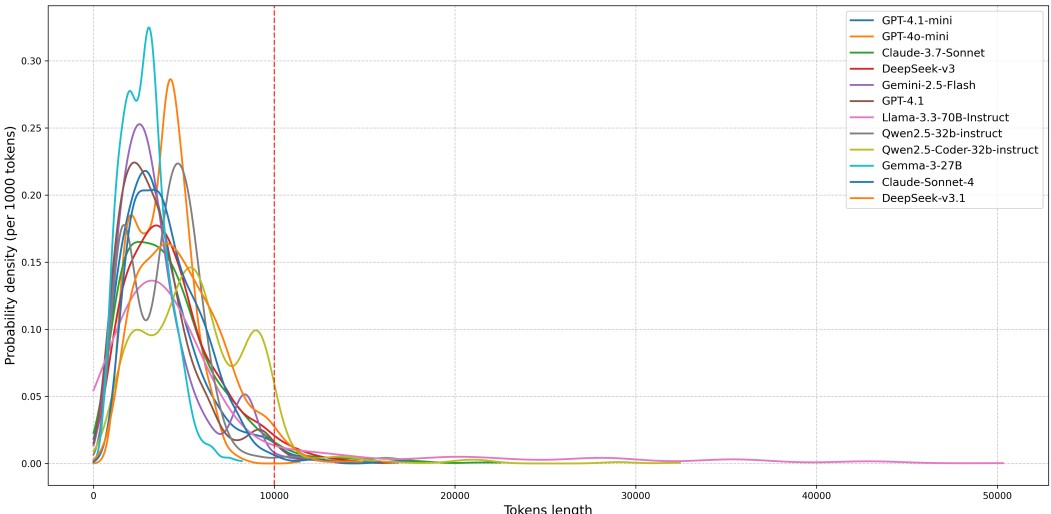

Figure 11: Final round tokens distribution.

Table 5: Token count summary.

| Model | Turn0 | Turn1 | Turn2 | Turn3 | Turn4 | Turn5 | Total Tokens |
|---|---|---|---|---|---|---|---|
| GPT-4.1-mini | 2,893,405 (880) | 2,373,016 (873) | 1,821,892 (701) | 1,395,392 (652) | 1,001,049 (633) | 511,243 (624) | 9,995,997 |
| GPT-4o-mini | 2,882,244 (880) | 2,339,212 (879) | 1,767,672 (708) | 1,326,185 (644) | 927,609 (627) | 511,353 (616) | 9,754,275 |
| Claude 3.7 Sonnet | 3,323,630 (880) | 2,662,097 (843) | 2,035,051 (661) | 1,544,448 (588) | 1,076,544 (545) | 430,239 (514) | 11,072,009 |
| DeepSeek v3 | 3,111,537 (880) | 2,456,677 (823) | 1,829,241 (649) | 1,351,863 (551) | 973,747 (518) | 415,113 (498) | 10,138,178 |
| Gemini 2.5 Flash | 2,320,363 (880) | 1,828,190 (818) | 1,360,611 (610) | 1,045,799 (563) | 767,764 (550) | 442,725 (540) | 7,765,452 |
| GPT-4.1 | 2,473,299 (880) | 1,990,849 (861) | 1,466,476 (640) | 1,105,167 (588) | 785,463 (568) | 467,807 (561) | 8,289,061 |
| Llama 3.3 70B | 4,092,643 (880) | 2,926,704 (880) | 1,948,855 (688) | 1,300,960 (600) | 858,866 (561) | 444,016 (536) | 11,572,044 |
| Qwen 2.5 32B | 2,851,011 (880) | 2,174,289 (880) | 1,527,872 (679) | 1,109,153 (617) | 771,020 (601) | 492,886 (594) | 8,926,231 |
| Qwen 2.5 Coder 32B | 3,027,046 (880) | 2,191,476 (714) | 1,575,748 (560) | 1,137,660 (520) | 765,829 (499) | 392,555 (489) | 9,090,314 |
| Gemma3 27B | 1,970,598 (880) | 1,495,326 (730) | 1,099,774 (575) | 825,266 (535) | 592,059 (513) | 412,435 (503) | 6,395,458 |
| Claude-Sonnet-4 | 3,163,255 (880) | 2,505,647 (880) | 1,866,589 (645) | 1,412,178 (581) | 1,001,346 (546) | 432,088 (522) | 10,381,103 |
| DeepSeek-v3.1 | 3,834,712 (880) | 3,072,138 (875) | 2,401,579 (689) | 1,899,652 (612) | 1,452,262 (579) | 459,179 (554) | 13,119,522 |

quality across different instruction difficulty levels, as well as examine instruction completion rates based on varying code difficulty levels. The results are shown in the Figure 12, 13, 14.

Considering the relationship between code quality and instruction difficulty in models, we found that the difficulty of instructions (4, 6, or 8 instructions) has a significant impact on the quality of code generated by most models. However, GPT-4.1 is an exception. We observed that GPT-4.1 is minimally affected by instruction difficulty, indicating that GPT-4.1 is well-suited for such tasks.

We observed no clear connection between code problem difficulty and instruction-following effectiveness, as the Figure 14 shows this metric seems largely unrelated to model performance or capability.

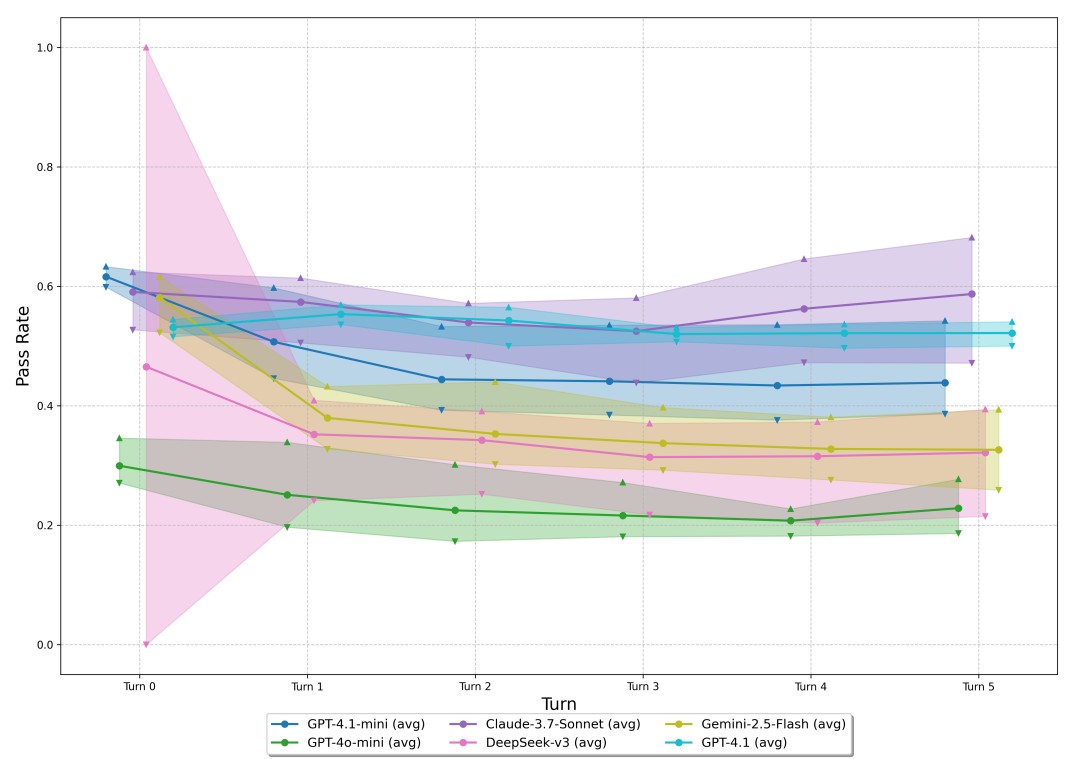

Figure 12: The Impact of instruction difficulty on pass@1 varies across different models (Group1).

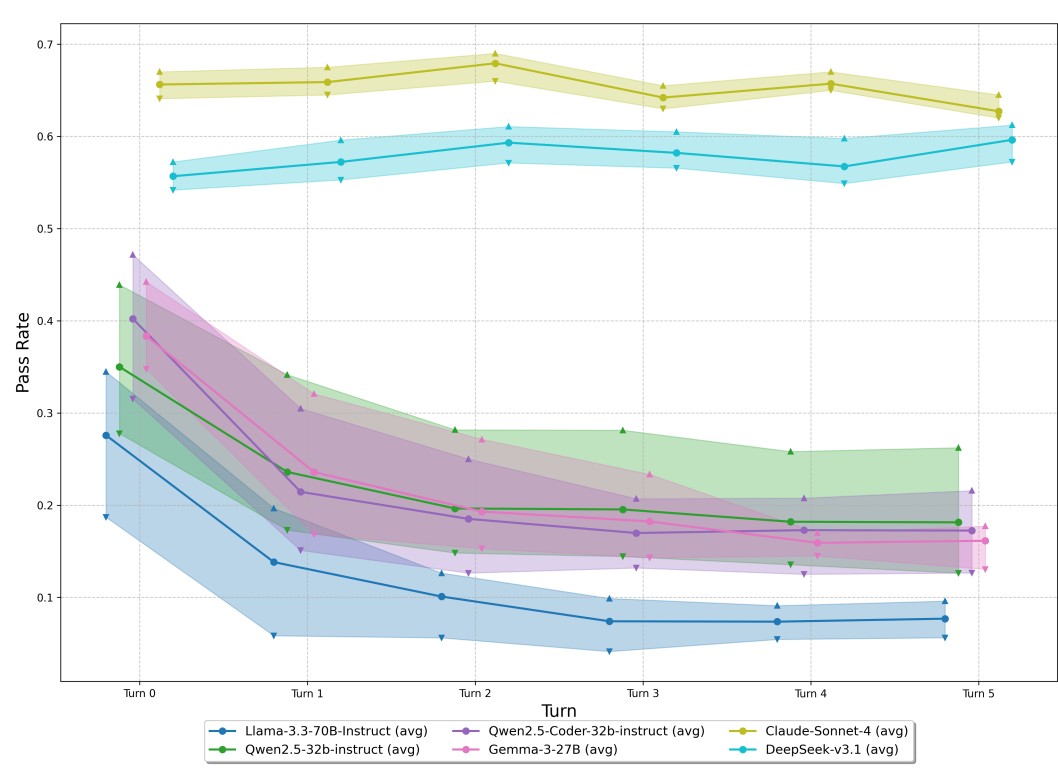

Figure 13: The impact of instruction difficulty on pass@1 varies across different models (Group2).

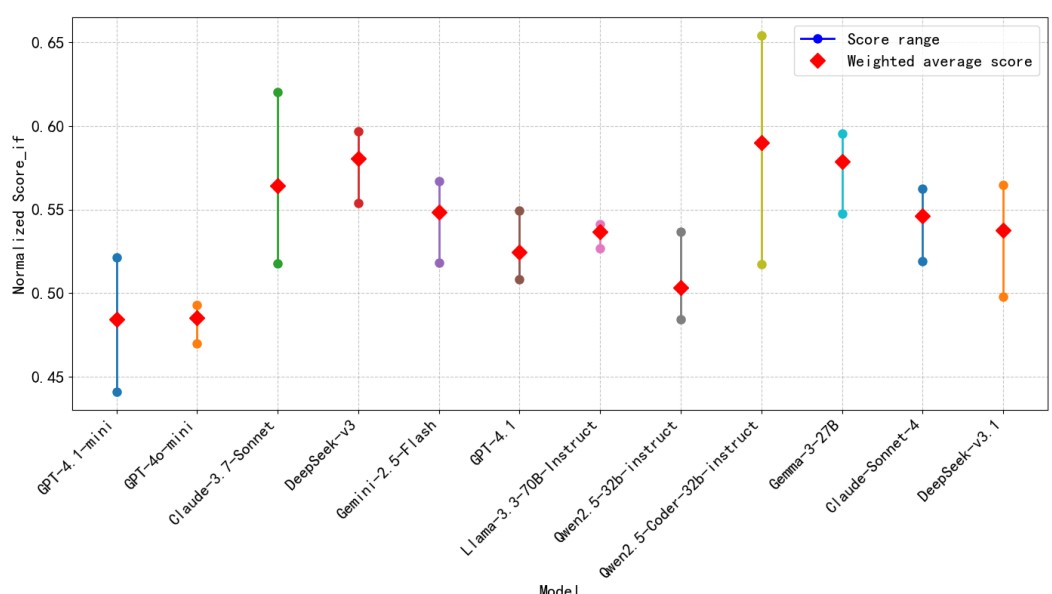

Figure 14: The standardized Score_if of different models is influenced by the difficulty of the coding problem.

### A.7 LLMs usage in our work

We hereby pledge that, regarding the use of LLMs in this work, we only used them for minor text refinement, apart from their evaluation purposes in the experiments.

### A.8 Prompts in our Experiments

Here we provide examples of the prompts used to generate baseline solutions and the experiment prompts.

1080
1081
1082
1083
1084
1085
1086
1087
1088
1089
1090
1091
1092
1093
1094
1095
1096
1097
1098
1099
1100
1101
1102
1103
1104
1105
1106
1107
1108
1109
1110
1111
1112
1113
1114
1115
1116
1117
1118
1119
1120
1121
1122
1123
1124
1125
1126
1127
1128
1129
1130
1131
1132
1133

**Baseline Solution Generating Prompt**

**Prompt:**
**System:** You are an expert Python programmer. You will be given a question (problem specification) and will generate a correct Python program that matches the specification and passes all tests. Your code should not only handle the test cases but also any possible input. The test cases are merely provided to facilitate your understanding of the problem.
**User:**
Problem: 2UP3DOWN
Takahashi is in a building with 100 floors. He uses the stairs for moving up two floors or less or moving down three floors or less, and uses the elevator otherwise. Does he use the stairs to move from floor X to floor Y?
Input:
The input is given from Standard Input in the following format: X Y
Output:
If Takahashi uses the stairs for the move, print Yes; if he uses the elevator, print No.
Constraints
- $1 \leq X, Y \leq 100$
- $X \neq Y$
- All input values are integers.
Sample Input 1:
1 4
Sample Output 1:
No
The move from floor 1 to floor 4 involves going up three floors, so Takahashi uses the elevator.
Sample Input 2:
99 96
Sample Output 2:
Yes
The move from floor 99 to floor 96 involves going down three floors, so Takahashi uses the stairs.
Sample Input 3:
100 1
Sample Output 3:
No
Test Cases:
Input 1: 1 4
Output 1: No
Input 2: 99 96
Output 2: Yes
Input 3: 100 1
Output 3: No
Please write code to solve this problem.
Important requirements: 1. Write a COMPLETE Python program, not just a function. 2. Include proper input handling code to read data directly (don't assume variables are pre-defined). 3. Call your main function directly at the end of the program. 4. Include all necessary imports. 5. The program should be ready to run without any modifications.
The test cases are provided to help you understand the problem, but your solution must work for all valid inputs.

**Experiment Prompt Example**

**Prompt:**
**System:** You are an expert Python programmer. You will be given a question (problem specification) and will generate a correct Python program that matches the specification and passes all tests. Your code should not only handle the test cases but also any possible input. The test cases are merely provided to facilitate your understanding of the problem.
**User:**
Problem: 2UP3DOWN
Takahashi is in a building with 100 floors. He uses the stairs for moving up two floors or less or moving down three floors or less, and uses the elevator otherwise. Does he use the stairs to move from floor X to floor Y?
Input:
The input is given from Standard Input in the following format: X Y
Output:
If Takahashi uses the stairs for the move, print Yes; if he uses the elevator, print No.
Constraints
- $1 \leq X, Y \leq 100$
- $X \neq Y$
- All input values are integers.
Sample Input 1:
1 4
Sample Output 1:
No
The move from floor 1 to floor 4 involves going up three floors, so Takahashi uses the elevator.
Sample Input 2:
99 96
Sample Output 2:
Yes
The move from floor 99 to floor 96 involves going down three floors, so Takahashi uses the stairs.
Sample Input 3:
100 1
Sample Output 3:
No
Test Cases:
Input 1: 1 4
Output 1: No
Input 2: 99 96
Output 2: Yes
Input 3: 100 1
Output 3: No
Please write code to solve this problem.
Important requirements: 1. Write a COMPLETE Python program, not just a function. 2. Include proper input handling code to read data directly (don't assume variables are pre-defined). 3. Call your main function directly at the end of the program. 4. Include all necessary imports. 5. The program should be ready to run without any modifications.
The test cases are provided to help you understand the problem, but your solution must work for all valid inputs.

**Experiment Prompt Example**

(continue)
**Assistant**:
Here is a complete Python program that solves the problem as described:

```python
def main():
    import sys

    # Read input from standard input
    input_data = sys.stdin.read().strip()
    X, Y = map(int, input_data.split())

    # Calculate the difference between the two floors
    difference = Y - X

    # Determine if Takahashi uses the stairs or the elevator
    if (1 <= difference <= 2) or (-3 <= difference <= -1):
        print("Yes")
    else:
        print("No")

if __name__ == "__main__":
    main()
```

Explanation:
1. **Input Handling**: The program reads input directly from standard input, which is suitable for competitive programming and other environments where input is provided in this manner.
2. **Logic**: It calculates the difference between the two floors (Y - X) and checks if this difference falls within the ranges that allow the use of stairs: - Moving up 1 or 2 floors (difference of 1 or 2). - Moving down 1, 2, or 3 floors (difference of -1, -2, or -3).
3. **Output**: It prints "Yes" if Takahashi can use the stairs and "No" otherwise.
4. **Execution**: The 'main()' function is called at the end of the program to ensure it runs when the script is executed.
This program is ready to run and will handle all valid inputs as specified in the problem constraints.

**User**:
Please revise the provided code to meet the following requirements:
1. Please revise your code to be written in Java.
2. Could you modify your code to ensure it contains at least one for loop?
3. It would be appreciated if you could adjust your code to run within 2629 milliseconds.
4. Please revise yourcode to exclude any function in the implementation.
5. We recommend updating your code to utilize item47 as a variable name.
6. It would be appreciated if you could adjust your code to make the 3rd variable a tuple.
7. All Your function should have at least 3 parameters.
8.We recommend refactoring your code to produce output in the [ output ] format.
And please remain other requirements in the previous prompt.

> ### Experiment Prompt Example
>
> (continue)
> **Assistant:**
> Here is the revised Java program that meets your requirements:
>
> ```java
> import java.util.Scanner;
>
> public class TwoUpThreeDown {
>     public static void main(String[] args) {
>         Scanner scanner = new Scanner(System.in);
>
>         // Read input values
>         int X = scanner.nextInt();
>         int Y = scanner.nextInt();
>
>         // Create a tuple-like structure using an array
>         int[] item47 = new int[2];
>         item47[0] = X;
>         item47[1] = Y;
>
>         // Calculate the difference
>         int difference = item47[1] - item47[0];
>
>
>         String result = "No"; // Default to "No"
>         for (int i = 0; i < 1; i++) {
>             if ((1 <= difference && difference <= 2) || (-3 <=
>             difference && difference <= -1)) {
>                 result = "Yes";
>             }
>         }
>
>         // Output the result in the specified format
>         System.out.println("[ " + result + " ]");
>
>         scanner.close();
>     }
> }
> ```
>
> Explanation of Changes:
> 1. **No Functions**: The code is structured without any additional functions, as per your request.
> 2. **Tuple-like Structure**: Instead of using a tuple, an array named 'item47' is used to hold the two floor values, simulating a tuple.
> 3. **For Loop**: A for loop is included to satisfy the requirement, although it only iterates once.
> 4. **Output Format**: The output is formatted as '[ result ]' to match your specifications.
> 5. **Performance**: The code is simple and efficient, ensuring it runs well within the time limit of 2629 milliseconds.
> This Java program is ready to run and will handle all valid inputs as specified in the problem constraints.

