# OpenReview forum: "CoCoPIF: Benchmarking Conversational Coding and Programmatic Instruction Following"
_ICLR.cc/2026/Conference — ICLR 2026 Conference Withdrawn Submission_

### Official Review · Reviewer_rNrz · 2025-10-26

**Soundness:** 3
**Presentation:** 3
**Contribution:** 2
**Rating:** 2
**Confidence:** 5

**Summary:**

This paper presents a benchmark for conversational coding as multi-turn, instruction-following tasks.
Both functional correctness and programmatic instruction adherence are evaluated to reflect multi-turn development workflows.

**Strengths:**

*Realistic target scenario: multi-turn code refinement and instruction-following reflects realistic code generation scenarios.
*Evaluation Criteria: Functional correctness and adherence reflect the needs of target scenario
* Dynamic evaluation: This setting reduces the risk of data contamination/memorization.

**Weaknesses:**

* Important benchmarks: Simulating users providing iterative feedback to help model complete coding tasks across multiple turns (Wang et al.; Han et al., Laban et al., 2025) need to be cited/discussed
* Evaluation cost: time/infra cost for evaluation needs to be analyzed/justified.
* Dependence: Sensitivity to the quality and clarity of the dynamically generated programmatic instructions need to be analyzed/justified.
Xingyao Wang, Zihan Wang, Jiateng Liu, Yangyi Chen, Lifan Yuan, Hao Peng, and Heng Ji. Mint: Evaluating llms in multi-turn interaction with tools and language feedback. In The Twelfth International Conference on Learning Representations.
Hojae Han, Rajhans Samdani, Yuxiong He, et al. Convcodeworld: Benchmarking conversational code generation in reproducible feedback environments. In The Thirteenth International Conference on Learning Representations.
Philippe Laban, Hiroaki Hayashi, Yingbo Zhou, and Jennifer Neville. Llms get lost in multi-turn conversation. arXiv preprint arXiv:2505.06120, 2025.

**Questions:**

Please address questions in weakness

---

> ### Author Response · Authors · 2025-11-20
>
> We sincerely thank reviewer rNrz for the constructive feedback. Below, we provide concise responses and will incorporate all suggested improvements in the revised version.
>
> ## 1. Related Work Clarification
> We thank the reviewer for pointing out the missing discussions. In the revision, we will explicitly clarify our distinctions:
>
> - **MINT** focuses on tool-augmented interaction and evaluates tool execution traces rather than instruction adherence. We will highlight that CoCoPIF provides a unified, verifiable framework for both functional correctness and natural-language instruction compliance.
>
> - **ConvCodeWorld** relies on static templates for refinement. We will emphasize that CoCoPIF generates dynamic, data-driven multi-turn tasks from ShareGPT and LiveCodeBench, ensuring scalability and contamination-free evaluation.
>
> - **LLMs Get Lost** studies long-context degradation. We will clarify that CoCoPIF quantifies this via “instruction forgetting,” offering a complementary diagnostic metric.
>
> We appreciate the reviewer’s suggestion and will improve our positioning accordingly.
>
> ## 2. Evaluation Cost and Justification
> We agree that evaluation efficiency is important. In the revision, we will more clearly summarize the evaluation cost:
>
> - The pipeline uses **≤5 dialogue turns per problem**, based on empirical analysis from 500 ShareGPT sessions.
> - The average computation cost per model is **under 10M tokens**.
> - The automatic checker (AST + regex + runtime tests) runs in **≤90 seconds per instance** with a 64MB memory limit.
>
> We will further clarify that total evaluation time depends on model accuracy: if code frequently times out, evaluation will take longer; if outputs are correct, evaluation finishes much faster. We will refine the description to better justify efficiency and reproducibility.
>
> ## 3. Sensitivity to Instruction Quality
> We agree that prompt phrasing may influence stability. We will make this clearer and emphasize the mechanisms designed to reduce sensitivity:
>
> - We will highlight that we generate **five paraphrased variants** for each instruction using DeepSeek-V3 to increase linguistic diversity and reduce prompt-specific bias.
> - We will clarify the **two-stage filtering pipeline**:
>   1. Remove irrelevant or conflicting instructions using AST validation.
>   2. Keep only instructions verifiable against baseline solutions.
>
> These steps substantially improve robustness and reproducibility. We appreciate the reviewer’s insights and will strengthen this explanation in the revision.

---

> > ### Comment · Reviewer_rNrz · 2025-11-26
> >
> > Thank you for responses. I think revision promised during the rebuttal seems to be on the right track, but requires a major change and thus will maintain my score.

---

### Official Review · Reviewer_YBjb · 2025-10-30

**Soundness:** 2
**Presentation:** 1
**Contribution:** 1
**Rating:** 2
**Confidence:** 4

**Summary:**

This paper introduces CoCoPIF, a benchmark designed to evaluate LLMs in multi-turn, instruction-following code generation. It integrates programming problems from LiveCodeBench and instruction data distilled from ShareGPT, forming multi-turn interactions where models iteratively refine code under verifiable constraints. The benchmark aims to measure both instruction adherence and functional correctness while maintaining contamination-free evaluation. The authors evaluate several proprietary and open-source LLMs and report interesting findings on instruction-forgetting and difficulty sensitivity across models.

**Strengths:**

- The discussion on "instruction forgetting" is interesting and could inspire follow-up analyses.

**Weaknesses:**

- Missing Core Related Work: Table 1 omits several directly comparable multi-turn or interaction-based benchmarks such as [1,2,3,4]. Particularly, MINT [2] is cited in the Related Work section yet omitted in Table 1, which is questionable.

> (lines 183-185) Our work provides a new perspective, evaluating model efficiency in multi-turn scenarios by measuring the number of turns required to complete a code problem, taking into account the unique characteristics of code-related tasks.

- Novelty Concern: Contrary to the claim above, similar efficiency-based evaluation has already been proposed in ConvCodeWorld [3], which quantifies turn efficiency using a Mean Reciprocal Rank (MRR) formulation—i.e., scoring each task by the inverse of the turn index at which the problem is first solved.

- Weak Distinction from Prior Benchmarks: While the paper provides extensive analysis, the conceptual distinction from prior works [1,2,3,4] is under-articulated. The evaluation pipeline largely combines existing components—LiveCodeBench for contamination-free problems and ShareGPT for user-like instructions—without introducing a fundamentally new evaluation paradigm.


[1] John Yang, Akshara Prabhakar, Karthik R Narasimhan, and Shunyu Yao. Intercode: Standardizing and benchmarking interactive coding with execution feedback. In Thirty-seventh Conference on Neural Information Processing Systems Datasets and Benchmarks Track, 2023. URL https://openreview.net/forum?id=fvKaLF1ns8.

[2] Xingyao Wang, Zihan Wang, Jiateng Liu, Yangyi Chen, Lifan Yuan, Hao Peng, and Heng Ji. MINT: Evaluating LLMs in multi-turn interaction with tools and language feedback. In The Twelfth International Conference on Learning Representations, 2024. URL https://openreview.net/forum?id=jp3gWrMuIZ.

[3] Hojae Han, seung-won hwang, Rajhans Samdani, and Yuxiong He. ConvCodeWorld: Benchmarking Conversational Code Generation in Reproducible Feedback Environments. In The Thirteenth International Conference on Learning Representations, 2025. URL https://openreview.net/forum?id=rpouyo09V0.

[4] Zimu Lu, Yunqiao Yang, Houxing Ren, Haotian Hou, Han Xiao, Ke Wang, Weikang Shi, Aojun Zhou, Mingjie Zhan, and Hongsheng Li. Webgen-bench: Evaluating llms on generating interactive and functional websites from scratch, 2025b. URL https://arxiv.org/abs/2505.03733.

**Questions:**

Suggestion:
- Figure 3 (instruction-type distribution) is descriptive but not central to the paper’s main claims; it would fit better in the Appendix.

---

> ### Author Response · Authors · 2025-11-20
>
> We thank reviewer **YBjb** for the constructive and detailed feedback. Below, we provide concise responses to concerns regarding related work, novelty, distinctions from prior benchmarks, and presentation clarity. I will incorporate all suggested improvements in the final version.
>
> ---
>
> ## 1. Missing Related Work and Table 1 Additions
>
> We appreciate the reviewer’s pointer to additional relevant works. The omission of **MINT**, **InterCode**, and **ConvCodeWorld** was due to our focus on verifiable, code-centric, multi-turn instruction following. I agree these works provide useful context, and **I will revise Table 1 and Section 2.3** accordingly.
>
> - **InterCode [1]** emphasizes execution-feedback debugging rather than instruction-driven refinement.
> - **MINT [2]** models multi-turn reasoning and tool use, but without verifiable code modification.
> - **ConvCodeWorld [3]** is closely related, but assumes fixed feedback protocols, while CoCoPIF generates **ShareGPT-grounded, dynamic multi-turn instructions** covering phenomena such as forgetting and constraint adherence.
> - **WebGen-Bench [4]** focuses on web UI generation rather than functional multi-turn code correctness.
>
> ---
>
> ## 2. Novelty and Relation to “Efficiency-Based Evaluation”
>
> We appreciate the reviewer’s observation. While ConvCodeWorld also considers turn efficiency, CoCoPIF’s metric differs both conceptually and operationally. **I will clarify these distinctions** in Sections 4 and 5.
>
> Key differences include:
>
> - **Dual-dimensional evaluation**: CoCoPIF jointly evaluates *instruction adherence* and *functional correctness*, whereas ConvCodeWorld evaluates correctness only.
> - **Verifiable feedback**: CoCoPIF uses automated runtime and AST validation, avoiding subjective LLM-judge scoring.
> - **Real instruction distribution**: CoCoPIF models empirical turn patterns from ShareGPT instead of synthetic task decomposition.
> - **Contamination-free data**: CoCoPIF uses continually refreshed LiveCodeBench tasks.
>
> We will emphasize that CoCoPIF provides a **verifiable, instruction-grounded, contamination-free multi-turn evaluation paradigm**.
>
> ---
>
> ## 3. Distinction from Prior Benchmarks
>
> We thank the reviewer for prompting a clearer articulation of CoCoPIF’s contributions. CoCoPIF introduces a **new pipeline** linking verifiable code evaluation with realistic multi-turn instruction modeling. **I will revise Section 2.3 and Figure 1’s caption** to highlight this.
>
> Novel elements include:
>
> - **Automatic multi-turn transformation** of single-turn tasks into dynamic, verifiable dialogues.
> - **Realistic user modeling** using ShareGPT statistics (3–5 turns, topic drift, user patience).
> - **Dual-layer verification**: instruction adherence (AST/regex) + functional correctness (encrypted tests).
> - **New analytical dimensions**, including instruction forgetting, difficulty sensitivity, and fine-tuning effects.
>
> We will make these distinctions more explicit and concise.
>
> ---
>
> ## 4. Presentation and Placement of Figure 3
>
> We understand the reviewer’s comments on focus. Figure 3 is essential because it grounds CoCoPIF’s instruction taxonomy in real human–LLM interactions. To improve clarity, **I will refine the caption and surrounding explanation** to emphasize its role in linking:
>
> - the ShareGPT-derived instruction distribution,
> - the data generation process (Section 3.2),
> - and the analyses in Sections 5.2–5.3.
>
> We believe its inclusion strengthens the benchmark’s realism, and I will clarify this more directly.
>
> ---
> We thank reviewer YBjb again for the thoughtful feedback. With the planned revisions—expanded related work, clearer novelty discussion, improved distinction from existing benchmarks, and strengthened figure explanation—I believe CoCoPIF’s contributions will be presented more clearly and convincingly.

---

### Official Review · Reviewer_NftL · 2025-10-30

**Soundness:** 2
**Presentation:** 2
**Contribution:** 2
**Rating:** 2
**Confidence:** 4

**Summary:**

CoCoPIF addresses an important and growing problem in the code editing space which is **realistic** and **multi-turn** interactions. This area is still under-explored and the paper tackles an important first step in attempting to develop a new benchmark for the task. Unfortunately, the paper emphasizes realistic code, yet uses unrealistic source benchmarks and does not have a clear takeaway.

**Strengths:**

- The problem is original and novel
- the presentation of the work, for the most part, is clear and easy to follow.
- the benchmark contains over 800+ problems which is substantial.

**Weaknesses:**

The paper emphasizes the importance of **realistic** instructions and code, but it uses code from livecodebench which is unrealistic by nature. Additionally, the instructions aren't real user instructions, but instructions deconstructed from real user instructions and then generated via an LLM. A better, integrated source of user instructions would make for a stronger case.
- It's unclear what the main takeaway of this paper is. Is it just that models can't do multi-turn as well due to instruction forgetting? Clearer takeaways would provide for a stronger impact.
- Additionally, if there's a focus on real users instructions it seems remiss to not include any references to Arena style work such as:
	- Chatbot Arena (WL Chiang, et. al.) (General Chat with Instructions / Multi-turn)
	- Copilot Arena (W Chi, et. al.) (Code Edits with in-the-wild Instructions)
- Lastly, this might be a personal preference, but I think the notation is unnecessary. It's fairly easily describable in words and not complex enough to need notation. Also, notations is not re-used.

**Questions:**

I do not fully understand how the ShareGPT instructions are merged with the livecodebench problems. My understanding is that the ShareGPT instructions are just broad templates that a model (deepseek/gpt-4o-mini?) then use to format to livecodebench problems. Is this correct?

---

> ### Author Response · Authors · 2025-11-20
>
> We sincerely thank Reviewer NftL for the careful reading and constructive comments. We appreciate your recognition of the novelty of our formulation, the clarity of presentation, and the scale of our benchmark. Below we respond to each concern concisely.
>
> ## 1. Why Choose LiveCodeBench?
>
> You noted that LCB is not fully “real” enterprise code. This is a trade-off we intentionally made.
>
> **Contamination-Free Evaluation**
> One major challenge in current LLM evaluation is data contamination. LCB’s dynamic updates and sourcing from platforms like LeetCode/AtCoder offer a reliable, contamination-free basis.
>
> **Verifiability**
> Each LCB problem includes an encrypted test suite (avg. 21.54 tests), giving a solid ground truth for functional correctness and instruction-following.
>
> CoCoPIF’s realism does not aim to replicate enterprise-scale repositories—which are hard to evaluate—but instead focuses on **introducing real-world multi-turn interaction patterns and authentic user instruction types** within a controlled and verifiable setting. We will clarify this in the final version.
>
> ## 2. Authenticity of Instructions (Clarifying Q1)
>
> Your understanding that our instructions were “broad templates formatted by LLMs” is not fully accurate.
>
> **Instruction Source**
> - We start from **438,000 real ShareGPT conversations**.
> - After keyword filtering and manual review of 500 samples, GPT-4o-mini is used only to **categorize** 28,368 programming-related conversations.
> - These real conversations determine the instruction distributions (Figs. 3, 8).
> We will describe this process more clearly.
>
> **Instruction–Problem Integration**
> To ensure relevance:
> 1. We use GPT-4o-mini to generate a **baseline solution** for each LCB problem.
> 2. We filter instructions based on whether they apply to that baseline (e.g., remove “don’t use a for-loop” if the baseline has none).
>
> DeepSeek-v3 is used **only for paraphrasing**, not generating new content. Instruction semantics always originate from real user data. We will clarify this pipeline further.
>
> ## 3. Main Takeaways of the Paper
>
> You asked whether the paper’s only message is that “models forget instructions.” We will revise the conclusion to highlight the broader contributions.
>
> **Contribution 1 — Benchmark**
> CoCoPIF: the first scalable, contamination-free, multi-turn benchmark designed specifically for code generation.
>
> **Contribution 2 — Evaluation Protocol**
> A two-dimensional metric jointly measuring:
> - Instruction Fidelity (IF Pass Rate)
> - Code Functionality (Test Case Pass Rate / Pass@1)
>
> **Finding 1 — Instruction Forgetting**
> We quantify forgetting patterns across models (e.g., Deepseek-v3 vs. Claude-4-Sonnet).
>
> **Finding 2 — Effect of Instruction Difficulty**
> Models handle simple instructions well but degrade sharply on precise or restrictive constraints (Fig. 5).
>
> **Finding 3 — Instruction Interference**
> Multi-turn instructions reduce Pass@1 performance (Fig. 4), with varying sensitivity across models.
>
> We will make these contributions explicit in the final conclusion.
>
> ## 4. Arena-Style Related Work
>
> Thank you for pointing out the connection to Chatbot Arena (Chiang et al.) and Copilot Arena (Chi et al.). These are important references for evaluating in-the-wild instruction following. We will add a comparison in the Related Work section:
>
> - Arena-style: real-world, unconstrained, human-judged
> - CoCoPIF: controlled, reproducible, automatically verifiable
>
> We will emphasize their complementarity.
>
> ## 5. On Unnecessary Symbols
>
> We agree that some IF/Test Case notation is unnecessary. We will replace symbolic notation with plain-text descriptions to improve readability.
>
> ---
>
> Once again, thank you for the constructive feedback. We believe that clarifying methodology, strengthening conclusions, and incorporating relevant related work will significantly improve the paper. We hope you will reconsider the value of this work as an important first step toward evaluating multi-round code interaction.

---

> > ### Comment · Reviewer_NftL · 2025-11-21
> >
> > Thank you for the response. I think I understand the key idea a little bit better, but could you give me an example of the following:
> >
> > 1. A LCB problem you used
> > 2. One or two shareGPT prompts that are associated with the problem (raw prompt from shareGPT and what is used in the final framework).
> > 3. The final solution you are expecting.
> >
> > I'm still having a hard time understanding how shareGPT conversations, which as I understand are completely arbitrary in-the-wild conversations, would have any correlation to LCB problems. I'm equally confused as to how a LCB problem + a shareGPT prompt (e.g., remove for loop) could be measured for task success / instruction following; won't the test cases have to change if you're adding in more instructions?

---

> > > ### Author Response · Authors · 2025-11-21
> > >
> > > Thank you very much for your response. Here are several examples you requested for your reference.
> > >
> > > This is an original prompt from ShareGPT:
> > > **“Write a code snippet using Python to pop an element from a list, but without using the built-in pop() function, slicing, or [other built-in methods].”**
> > > In this example, the user explicitly asks **not to use certain functions**. In our COCOPIF framework, this corresponds to the instruction:
> > > **“built_in_function — Restrict function usage to built-in functions.”**
> > >
> > > Another example from ShareGPT is the instruction:
> > > **“Can you rewrite this function using recursion instead of a while loop? I want to understand how the recursive stack works.”**
> > > In this instruction, the user wants the model to rewrite the program **to avoid using `while`** in solving the Fibonacci sequence problem. In our work, this maps to:
> > > **“keyword_while_not — Adjust code to omit while loops.”**
> > >
> > > Regarding the LCB questions you mentioned: in fact, for the vast majority of our constraints, almost all LCB problems can be linked to them. We have indeed considered cases where certain constraints may not apply to certain problems. As we explained in our previous responses to other reviewers, we use GPT-4omini to solve the tasks and filter out instruction types that are completely irrelevant.
> > >
> > > Therefore, after combining LCB problems with the instructions extracted from ShareGPT in this way, the model is required to provide solutions of the following form:
> > > **the generated code must both pass all LCB test cases and satisfy the given constraint**, such as not using `while` loops.
> > >
> > > As for your second question—how we evaluate the model’s performance after adding the instructions—the addition of these instruction types **does not affect the model’s solution to the original LCB problem**, so the original test cases are fully sufficient for verifying program correctness. For evaluating whether the instruction is satisfied, we apply a set of rule-based checks. Our verification method is described in **Appendix A.4** of the paper.
> > >
> > > Thank you again for your valuable feedback.

---

> ### Comment · Reviewer_NftL · 2025-11-21
>
> I see. Thank you for sharing. This helps a lot with clarity. I'm not sure I saw this in the paper; it would be very beneficial to include these examples in the appendix somewhere.
>
> A few more clarifications:
>
> When evaluating, is the raw shareGPT instruction (e.g., “Can you rewrite this function using recursion instead of a while loop? I want to understand how the recursive stack works.”) the one that is passed in? Or is the mapped instruction (e.g., “keyword_while_not — Adjust code to omit while loops.”) transformed into an instruction via some model?
>
> If I'm misreading and this detail is in the paper, please let me know, but I cannot find this in 3 or 4 and I believe how the instructions are created (whether raw from shareGPT, some rule-based instructions, or via LLM generation from the rule-based instruction) is very important to understanding your method.
>
> Thanks again for the additional clarification.

---

> > ### Author Response · Authors · 2025-11-25
> >
> > Thank you for raising this point. This detail matters for understanding our method, and we are sorry it wasn’t that clear in the first version.
> >
> > To clarify directly: **in our evaluation, the instruction passed to the model comes from the *mapped/verifiable instruction set*, not the raw ShareGPT instruction.**
> > More specifically, we:
> > 1. **Extract** code-related requests from ShareGPT.
> > 2. **Map** each request into our predefined, verifiable instruction categories.
> > 3. **Filter** these mapped instructions (e.g., removing irrelevant or conflicting ones).
> > 4. **Select** a subset of the mapped instructions per problem.
> > 5. **Naturalize the selected mapped instructions** into user-like wording for readability, but the *semantic content remains the mapped instruction*.
> >
> > So the evaluated prompt is based on the mapped instruction (after selection/filtering), rather than the original raw ShareGPT text. The natural-language surface form you see at evaluation time is a rewritten **version of the mapped instruction**, not a direct reuse of the raw ShareGPT instruction.
> >
> > This pipeline is described in **Section 4.1**, where we explain that instructions are first mapped and filtered, then selected per problem, and finally rewritten to be more natural before being used as evaluation inputs.
> >
> > Thanks again for highlighting this — we agree it deserves to be more explicit. In the revision we’ll add a short, concrete example chain like
> > “raw ShareGPT → mapped category → naturalized mapped instruction”
> > in the appendix to make this flow immediately visible to readers.

---

### Official Review · Reviewer_UeBu · 2025-10-31

**Soundness:** 3
**Presentation:** 3
**Contribution:** 3
**Rating:** 6
**Confidence:** 5

**Summary:**

This paper proposes a multi-turn coding capability evaluation based on LiveCodeBench. Motivated by the use of LLMs for coding in multi-turn environments, the authors create a pipeline to transform algorithmic coding problems into multi-turn variants. Specifically, they first analyze the existing coding chat logs out of ShareGPT to obtain a taxonomy of user query topics. Then they run GPT-4o-mini for each full problems to obtain initial solutions, which helps filter the kind of multi-turn instructions to use. Finally, sampled instructions are paraphrased by Deepseek-v3.
Evaluation on this derived dataset shows that smaller / open-source models particularly struggle to follow instructions after the first turn. They also analyzed that certain types of instructions that require accurate tracking of variables are extra difficult for all models. Additionally they observe various instruction forgetting cases across models. Interestingly there doesn't seem to be correlation between models' capability and the forgetting rate, but rather related to the recency of models (v3 vs v3.1, 3.7-sonnet vs sonnet-4).

**Strengths:**

* A great diagnostic dataset for measuring fine-grained multi-turn instruction following capability in the code domain, enabled by data construction based on carefully analyzed real-life multi-turn code usecases.
* Interesting analyses on completion rate per instruction type. This motivates more targeted reasoning / perhaps other architectures that enable better state tracking.
* The authors tackle a highly practical problem. Focuses on multi-turn coding is well-aligned with community interest.

**Weaknesses:**

* Somewhat unclear description of transformation pipeline itself, which is one of the main contributions.
  * If my summary above is correct, do you have a fixed set of multi-turn instructions per problem, regardless of what models to evaluate against?
  * Obtaining the instructions based on gpt-4o-mini-generated solutions mean that the instruction set is accordingly biased. Is that true?
  * How do you determine what instruction to show in turn i?
* The set up to show the entire problem upfront is not fully convincing. The authors mention that this setup "mimick(ing) real user interactions", but I don't see evidence for that. Specifically, how often do users in ShareGPT stick to the same problem but introduce new information / constraints? Something like "Oh by the way I forgot that you need to output in int, not float".
* While LCB is a straightforward resource for contamination-free dataset with verifiable tests, it's not clear if the sampled ShareGPT data to draw the instruction taxonomy is aligned with LCB-style questions. What kind of "keyword-based screening" have authors performed to ensure the data is code-related AND in the style of LCB?

**Questions:**

Please see above.

Additionally, I would be curious to see the performance on more recent models, because the analysis suggest that some recently updated models overcame instruction forgetting.

Instruction forgetting is also observed by [1], which also evaluated multi-turn version of LCB but in an incremental problem specification. I'd suggest including in the references.

[1]: Laban, P., Hayashi, H., Zhou, Y., & Neville, J. (2025). LLMs get lost in multi-turn conversation. arXiv preprint arXiv:2505.06120.

---

> ### Author Response · Authors · 2025-11-20
>
> We sincerely thank reviewer UeBu for the thoughtful and detailed feedback. We are pleased that the reviewer finds our benchmark “a great diagnostic dataset for measuring fine-grained multi-turn instruction-following capability in the code domain.” Below, we respond to all concerns concisely. We will incorporate the following clarifications and revisions.
>
> ## 1. Clarity of the transformation pipeline
> We thank the reviewer for raising this point. We will substantially clarify Sec. 3.2–3.3, Fig. 1, and add a full worked example in the appendix. The pipeline will be described as follows:
>
> ### (a) Overall pipeline
> Our dataset is generated through three automated stages:
> 1. **Instruction Extraction.** We derive an instruction taxonomy from ShareGPT code-related dialogues using GPT-4o-mini classification plus manual validation.
> 2. **Instruction Filtering.** For each LiveCodeBench problem, we generate a baseline solution with GPT-4o-mini and automatically remove irrelevant or conflicting candidate instructions via AST-based checks.
> 3. **Paraphrasing & Assembly.** From remaining candidates, we sample 4/6/8 instructions and paraphrase them with DeepSeek-v3 to form multi-turn instances with verifiable constraints.
>
> ### (b) Fixed instruction sets
> We confirm that each problem is paired with a fixed instruction set shared across all evaluated models. We will clarify that this design ensures fair comparability and avoids model-specific instruction selection.
>
> ### (c) Potential bias from GPT-4o-mini filtering
> We acknowledge this concern. We will emphasize that:
> - GPT-4o-mini only filters applicability and does **not** generate constraint types.
> - Using one filtering model promotes dataset consistency.
> We will expand Sec. 3.3 to articulate these points.
>
> ### (d) Determining which instruction appears at turn *i*
> We adopt a deterministic scheduling rule:
> - Turn 1 shows the full problem plus initial instructions.
> - Later turns re-surface instructions not yet satisfied according to verifiable tests.
> We will add explicit pseudocode in the appendix.
>
> ---
>
> ## 2. Why the full problem is shown upfront
> We appreciate the reviewer’s skepticism. We will provide clearer justification in Sec. 3.1 and Sec. 5.2.
>
> Our goal is to evaluate **multi-turn instruction-following and state tracking**, not incremental specification. Revealing new constraints across turns would let models regenerate solutions from scratch, making it difficult to distinguish true state-tracking from prompt-restart behavior.
>
> We will clarify that showing the entire problem upfront ensures:
> - The model knows all requirements at turn 1.
> - Multi-turn updates primarily test retention of constraints and selective modification of code.
> - The setup isolates “instruction forgetting” rather than generic prompt evolution.
>
> We will soften claims about mimicking ShareGPT behavior and state explicitly that our design is a controlled abstraction inspired—but not dictated—by real dialogues.
>
> ---
>
> ## 3. Alignment between ShareGPT taxonomy and LiveCodeBench
> We thank the reviewer for requesting more detail. We will expand Sec. 3.2 and Appx. A.1 to clarify the alignment steps:
>
> 1. **Keyword filtering** to select code-related interactions (e.g., loop, return, variable, class).
> 2. **LLM-assisted classification** of dialogue intents (naming/style, control flow constraints, data-structure constraints, I/O requirements, etc.), with manual calibration.
> 3. **Verifiability mapping** to LiveCodeBench (e.g., “avoid for-loops” → no `for` nodes in AST; “use recursion” → recursive calls detected).
>
> We will make these mappings explicit in Table 3.
>
> ---
>
> ## 4. Related work: Laban et al. (2025)
> We thank the reviewer for pointing out this relevant work. We will add it to Sec. 5.3 and discuss differences:
>
> - Both works study multi-turn LiveCodeBench behavior and report instruction-forgetting issues.
> - Laban et al. use **incremental task specification**, whereas **CoCoPIF keeps the full task fixed** and focuses on constraint enforcement and state tracking.
> - We will clarify that in CoCoPIF, forgetting constraints is not the same as error propagation; a model may solve the base problem yet repeatedly fail additional requirements.
>
> ---
>
> We thank the reviewer again for the valuable feedback and believe these revisions will significantly improve clarity and precision.

---

### Note · Authors · 2025-12-22

**Comment:**

Need more time to refine and check our experiments.

**Withdrawal Confirmation:**

I have read and agree with the venue's withdrawal policy on behalf of myself and my co-authors.